# Neurovascular Unit Compensation from Adjacent Level May Contribute to Spontaneous Functional Recovery in Experimental Cervical Spondylotic Myelopathy

**DOI:** 10.3390/ijms24043408

**Published:** 2023-02-08

**Authors:** Guang-Sheng Li, Guang-Hua Chen, Kang-Heng Wang, Xu-Xiang Wang, Xiao-Song Hu, Bo Wei, Yong Hu

**Affiliations:** 1Spinal Division of Orthopedic and Traumatology Center, The Affiliated Hospital of Guangdong Medical University, Zhanjiang 524002, China; 2Department of Orthopaedics and Traumatology, The University of Hong Kong, Hong Kong, China

**Keywords:** neurovascular unit compensation, chronic compressive spinal cord injury, cervical spondylotic myelopathy

## Abstract

The progression and remission of cervical spondylotic myelopathy (CSM) are quite unpredictable due to the ambiguous pathomechanisms. Spontaneous functional recovery (SFR) has been commonly implicated in the natural course of incomplete acute spinal cord injury (SCI), while the evidence and underlying pathomechanisms of neurovascular unit (NVU) compensation involved in SFR remains poorly understood in CSM. In this study, we investigate whether compensatory change of NVU, in particular in the adjacent level of the compressive epicenter, is involved in the natural course of SFR, using an established experimental CSM model. Chronic compression was created by an expandable water-absorbing polyurethane polymer at C5 level. Neurological function was dynamically assessed by BBB scoring and somatosensory evoked potential (SEP) up to 2 months. (Ultra)pathological features of NVUs were presented by histopathological and TEM examination. Quantitative analysis of regional vascular profile area/number (RVPA/RVPN) and neuroglial cells numbers were based on the specific EBA immunoreactivity and neuroglial biomarkers, respectively. Functional integrity of blood spinal cord barrier (BSCB) was detected by Evan blue extravasation test. Although destruction of the NVU, including disruption of the BSCB, neuronal degeneration and axon demyelination, as well as dramatic neuroglia reaction, were found in the compressive epicenter and spontaneous locomotor and sensory function recovery were verified in the modeling rats. In particular, restoration of BSCB permeability and an evident increase in RVPA with wrapping proliferated astrocytic endfeet in gray matter and neuron survival and synaptic plasticity were confirmed in the adjacent level. TEM findings also proved ultrastructural restoration of the NVU. Thus, NVU compensation changes in the adjacent level may be one of the essential pathomechanisms of SFR in CSM, which could be a promising endogenous target for neurorestoration.

## 1. Introduction

Cervical Spondylotic Myelopathy (CSM) is the most common cervical spinal cord dysfunction in middle-aged and elderly people. Vascular dysfunction and neural impairment have been demonstrated to be a crucial pathophysiological process in chronic compressive SCI [1,2]. Surrounding degenerative tissue, inducing intermittent or continuous compression to the spinal cord, which cause ischemia, disruption of blood spinal cord barrier (BSCB), neuronal apoptosis and damage of descending or ascending axon, eventually lead to incomplete neurological deficit [3,4]. Although it has been widely believed that neural regeneration is extremely limited in complete SCI of adult mammals, spontaneous functional recovery (SFR) commonly occurs in incomplete SCI [5,6,7]. Restoration of BSCB integrity, neuroprotective effects by neurotrophic factor secretion, axonal remyelination by oligodendrocyte precursors and axonal remodeling such as axonal sprout formation may contribute to spontaneous recovery after incomplete SCI [7,8].

The neurovascular unit (NVU), integrated by neurons, microvascular endothelial cells and the surrounding neuroglial cells, i.e., oligodendrocytes and astrocytes, constitutes a fundamental functional unit to enable intercellular communication and signaling transduction [9,10]. The highly specified monolayer endothelial cells with the perivascular pericytes and the surrounding astrocytes (astrocytic endfeet) constitute the integrated BSCB, which plays an essential role in regulating blood supply and substances exchange, thus maintaining microenvironment homeostasis of CNS [11,12]. The osculating connection facilitates cell to cell interaction among NVU compartments. Interactions between endothelial cells and astrocytes maintain the functional integrity of the BSCB [11]. Therefore, it is necessary to reveal the association change among components of NVU and fully elucidate the pathophysiological mechanism of this association in the natural process of SFR, which could help to shed light on the promising therapeutic strategy of CSM.

Although numerous studies have demonstrated that dysfunction or disruption of NVU is the main pathophysiological change response for ischemic cerebral insult or degenerative central nervous system (CNS) disorders [10,13,14,15], the existing evidence regarding spontaneous NVU remodeling in the structural and functional recovery of SCI is extremely limited. A recent study has revealed several degenerative ultrastructural changes of NVU including degeneration of neurons and axons, vacuolation of endothelium, breakdown of the tight junction and swelling of astrocytic endfeet and mitochondria in chronic compressive cervical SCI [16]. Compensatory changes of vascular components including proliferation of endothelium, thickening of basement membrane and increment of pericyte vessel coverage were observed in the compressive epicenter [16]. However, the pathomechanism of NVU compensation contributing to SFR has not been elaborated.

As a specific marker for CNS blood vessels, endothelial barrier antigen (EBA) was used to detect the integrity of the BSCB [17,18]. The present study was an elementary attempt to disclose pathophysiological changes in each NVU component, in particular in the adjacent level, in an experimental CSM model. Ultrastructural features of NVU components were revealed by Transmission Electron Microscopy (TEM). Evan blue (EB) extravasation testing was used to detect the permeability of the BSCB. EBA-immunoreactivity was used for blood vessel quantitative analysis. Neuroglial cells including astrocytes, oligodendrocytes and microglial cells were also specifically visualized by immunohistochemical staining. The interrelation among quantity of microvessel and neuroglial cells were analyzed.

## 2. Results

### 2.1. Water-Absorbing Polymer Induced Chronic Compression to the Cervical Cord

The establishment of an experimental CSM model can be demonstrated with preoperative X-ray with the location of cervical spine level (Figure 1A), a view of the water-absorbing sheet on the left side of C5 dorsal epidural space without any visible bleeding or acute injury under operating microscope intraoperative (Figure 1B), post-operative specimen inside the spinal canal (Figure 1C) and ex-vivo (Figure 1D). Observation on later H&E and LFB results did not find without evident spinal cord edema, hemorrhage, intramedullary cavity and glial scar after dorsal–lateral cord compression.

### 2.2. SFR in Modeling Rats

SFR was testified by 21-point BBB scoring and somatosensory evoked potential (SEP). In the CSM group, significant paralysis of the left forelimb was observed but with a preferable forelimb–hindlimb coordination and consistent plantar stepping. BBB scoring showed significant decline during the period between 1st day and 7th day (Figure 2A), while SEP showed prolonged latency (Figure 2B) and decreased amplitude. (Figure 2C) From 1st month to 2nd month, the locomotors went through slow recovery. Meanwhile, there was significant recovery of SEP latencies and amplitudes at 2 months after compression, without significant difference in latency or amplitude between preoperative and 2-month compression test (*p* > 0.05).

### 2.3. Neuronal Degeneration and Axonal Demyelination after Compression

Basophilic appearance of neuron nuclei with rich cytoplasm and organized neural fiber with intact myelin sheath was seen in the control group. Spindle-shaped appearance, loss of cytoplasm and nucleus karyopyknosis and decreased neuronal synapses were the typical characteristic of the large motoneuron in the compressive epicenter, as well as the adjacent level (Figure 3A). Axonal demyelination change was verified by LFB staining. In posterior–lateral funiculus of the compressive epicenter, disorganized neural fibers and broken axon network with obvious vacuolation were observed (Figure 3B).

In a quantitative analysis, there is no significant reduction in the number of large motor neurons found in the ventral horn (VH) (*p* < 0.05) (Figure 3C). A significant reduction of blue staining intensity was found in the compressive epicenter compared to the control group (*p* < 0.05), but no significant difference in staining intensity (Figure 3D) was found between adjacent level and the control group (*p* > 0.05).

Ultrastructural findings by TEM further confirmed neuronal degeneration and axonal demyelination (Figure 4). In the control group, the Nissl body (Nb) was evenly distributed in the neuron nucleus and rich mitochondria were identified with distinguishable cristae (Figure 4(A1)). In the compressive epicenter, results showed high electronic density of the neuron nucleus and low density of the cytoplasm (Figure 4(A2)), as well as mitochondrial swelling and cristae disappearance (Figure 4(A2)) and thinner myelin and destruction of the neurofilament (Figure 4(B2)). Relative normal ultrastructure of neuron (Figure 4(A3)) and reorganization of axons and the myelin sheath (Figure 4(B3)) were seen in the adjacent level. Compared with the distinguished mitochondrial cristae of the synapse in the control group, loss of mitochondrial cristae and synaptic vesicle of synapses were observed in the compressive epicenter (Figure 4(C2)). In the adjacent level, a similar appearance of synapse was seen to that of the control group and the number of mitochondria and synaptic vesicles increased (Figure 4(C3)).

### 2.4. EBA Immunoreactivity and Anatomical Distribution of Microvessels

In the control group, all *microvessels* of gray matter (GM) and white matter (WM) were immunopositive (Figure 5A). The EBA located at the luminal surface of endothelial cells of capillary and pial vessels appeared to be golden brown. The pericytes embraced the endothelial cells and constituted the microvascular wall of BSCB and were stained in dark blue in the nucleus by hematoxylin counter stain. The profile of EBA-immunopositive longitudinal vascularization appeared to be a smooth uninterrupted tubular structure, while the transverse vascularization displayed as a ring sign (Figure 5A).

The regional vascular profile area/number (RVPA/RVPN) in GM accounted for 76.68% (63.92%), which was nearly twice that of WM at 23.32% (36.08%) (Figure 5B,C), which demonstrated rich blood supply in GM. The majority of blood vessels were distributed specifically the ventral horn (VH) of GM (Figure 5A). The VH, where large motor neurons are located, contained the largest proportion of RVPN (RVPA), at 49.29% (42.12%), compared with 27.39% (21.80%) in the dorsal horn (DH). In contrast, the blood vessels were sparse in WM, with only 8.83% (11.90%), 7.51% (17.33%) and 6.97% (6.85%) in anterior funiculus (AF), posterior funiculus (PF) and lateral funiculus (LF), respectively (Figure 5B,C). A significant difference of RVPN (RVPA) was found between GM and WM, VH and DH, VH and AF, VH and PF, VH and LF, DH and AF, DH and PF, DH and LF, while there was no significant difference among AF, PF and LF (Figure 5D,E). Additionally, strong linear relationships were observed between RVPN and RVPA. These findings suggested that the GM, especially the VH, was the most vascularize region of spinal cord, thus, provided abundant blood supply for neuronal metabolism activity.

### 2.5. BSCB Disruption in the Compressive Epicenter

The permeability of the BSCB was detected by the extravasated EB contained in the parenchyma of spinal cord tissue (Figure 6). EB fluorescence increased significantly in the compressive epicenter from the 7th day to 2nd month, as well as in the adjacent level at the 7th day (Figure 6A–E). The permeability disruption even occurred in the adjacent level at the early stage. The quantitative results by the optical density (OD) value of EB extravasation content further proved the fluorescence findings (Figure 6F). Endothelial barrier dysfunction was also demonstrated in the compressive epicenter by EBA immunostaining (Figure 5A). The vascular profile, labelled by the specific EBA, looked like a worm-eaten pattern, was completely destroyed, or even disappeared in the ipsilateral and contralateral cord (Figure 5A). In addition, quantitative analysis of blood vessels demonstrated significant loss of RVPN/ RVPA in the compressive epicenter.

### 2.6. Microvascular Compensation in Adjacent Level

TEM showed a significant decrease of endothelial vacuoles and tight junction defects, while the astrocytic endfeet area increased in the adjacent level (Figure 7(B3,C3),E,F). EBA-immunoreactivity reappeared and the vascular profile appeared to be an integral tubular morphology, resembling the control group, and the RVPN/RVPA increased significantly in the VH and DH of the adjacent level (Figure 5D,E). Furthermore, the perivascular astrocytic endfeet proliferated more conspicuously and encompassed the microvessel closer than that in the compressive epicenter (Figure 7(A1–A3)). The astrocytic endfeet coverage, specified by GFAP staining, also increased significantly (Figure 7D). The lamellar structure of the basement membrane returned to be easily distinguishable and the collapsed vascular contour was restored. Despite the permeability of the BSCB in the adjacent level, which could not be spared from disruption at 7th day, the integrity of barrier permeability was almost completely restored at the 2nd month (Figure 6).

### 2.7. Neuroglial Cells Morphology and Regional Distribution

In IHC counterstaining sections by Mayer’s hematoxylin, neuroglia can be identified by their feature nucleus. The ultrastructural feature presented as even distribution and low-density karyoplasm with condensed rim close to the karyolemma in the control group (Figure 8(A1)). The astrocyte has more abundant cytoplasm, containing cytoplasmic organelles such as granular endoplasmic reticulum, Golgi complexes, ribosomes and mitochondria. In the compressive epicenter, the chromatin of the nucleus densely clumping around the karyolemma was the most prominent difference (Figure 8(A2)), while similar findings were also identified in the adjacent level (Figure 8(A3)).

In the control group, cytoplasmic organelles such as granular endoplasmic reticulum, free polysome and mitochondria were usually seen, while mitochondria were spared (Figure 8(B1)). In the compressive epicenter, the chromatin was more clumped but endocentrically distributed. Denser cytoplasm, malformed nucleus, nucleocytoplasmic separation and cytoplasmic vacuolation was distinguished (Figure 8(B2)). However, oligodendrocytes in the adjacent level appeared to have normal morphology (Figure 8(B3)).

Microglial cells are often irregular in outline or have a small elongate cell body with clumped chromatin in the control group (Figure 8(C1)). The nucleus was also irregular in shape, typically with a rod-shape. In the compressive epicenter, some microglial cells were aggregated and closely in contact with compromised EBA-immunopositive microvessels. Other microglia expand as a foam cell or gitter cells which would have phagocytosed some cellular debris (Figure 8(C2)).

A large proportion of neuroglia cells, including oligodendrocytes, astrocytes and microglia cells, were mainly distributed in the GM, especially in the VH. Astrocytes were most abundant neuroglial cells in the spinal cord (Figure 8(D1–D3)). The oligodendrocytes number in the GM were almost equal to that in the WM (*p* > 0.05). Meanwhile, the number of astrocytes and microglial cells in the GM were about twice that in WM (*p* < 0.01). The microglial cells accounted for a small part of the neuroglia, while they were increased in the compressive epicenter and adjacent level. There was no significant difference in neuroglial cell quantity among AF, PF and LF regions of the WM.

### 2.8. Neuroglial Cells Activation after Chronic Compression

The specific immunostaining of astrocytes, oligodendrocytes and microglial cells showed evident activation after compression, especially in the compressive epicenter. Activation of astrocytes can be clearly identified by the immunoreactivity of GFAP antibodies and hematoxylin nucleus staining, which was characterized by a larger nucleus and cytoplasmic process gliosis (Figure 9(A2)). The reactive astrocytes were referred to as “gemistocytic astrocytes’’ or ‘‘gemistocytes’’. In the compressive group, a large number of activated astrocytes aggregated in the post column of the compressive epicenter (Figure 9(A2)). The astrocyte number increased significantly in the compressive epicenter, especially in the dorsal and ventral horn of the compressive epicenter (*p* < 0.05). Interestingly, a noticeable increment of astrocyte number was observed in the posterior funiculus of the adjacent level rather than in the compressive epicenter. In comparison with the compressive epicenter, mild reactive astrocytosis was seen in the adjacent level (Figure 9(A3)), with a larger number than that in the GM of control group (*p* < 0.05). The positive immunoreactivity oligodendrocyte showed a similar “fried egg” sign as the pathological feature, with a clear rim of nuclear and hypochromatic cytoplasm in the control group (Figure 9(B1)). In contrast, the specific marked cells appeared as a dense cytoplasm and the nuclear membrane became indistinct. There was a large number of cells aggregated in the GM and the posterolateral column of WM in the compressive epicenter (Figure 9(B2)) compared with the control group (*p* < 0.05). The activated cells were also found to accumulate in the adjacent level (Figure 9(B3)), while the findings were only statistically significant in the ventral horn and the lateral funiculus. Activation of microglial cells was marked by the Anti-Iba1 antibody (Figure 9(C1–C3)). The number of microglial cells increased dramatically in both GM and WM after compression (*p* < 0.01). Meanwhile, a significant increment of astrocyte number was observed in GM (*p* < 0.01).

### 2.9. Interrelations of Microvessels and Neuroglial Cells

A matrix scatter plot was used to disclose the interrelations between vessels (RVPN/RVPA) and neuroglial cells (number of oligodendrocytes, astrocytes and microglial cells) in the compressive epicenter (Figure 10). A strong positive correlation was found between RVPN and RVPA (r_vn-va_ = 0.830, *p* = 0.000). A moderate correlation was found between astrocyte number and RVPN/ RVPA (r_a-vn_ = 0.612, *p* = 0.000; r_a-va_ =0.472, *p* = 0.000), as well as the number of astrocytes and oligodendrocytes (r = 0.470, *p* = 0.000). Meanwhile, a weak correlation was seen between oligodendrocyte/microglial cell number and RVPN/ RVPA (r_o-vn_ = 0.226, *p* = 0.000; r_m-vn_ = 0.183, *p* = 0.010; r_o-va_ = 0.268, *p* = 0.000; r_m-va_ = 0.092, *p* = 0.019), microglial cell number and oligodendrocyte number and microglial cell number and astrocyte number (r = 0.156, *p* = 0.127; r = 0.294, *p* = 0.000).

## 3. Discussion

The clinical presentation of CSM varies with a big spectrum of spontaneous progression. Some patients with CSM showed linear deterioration in neurological function. Some patients present stable disability followed by episodes of deterioration, while others showed a steady disability to a sudden decline. Currently, surgical decompression is the mainstay treatment strategy of CSM. However, the efficacy of neurological recovery following surgical decompression appears to be a large uncertainty, i.e., some patients achieve satisfactory recovery, while others remain residually disabled, even showing unpredictable deterioration. The pathomechanism underlying such existing clinical contradictions remains poorly understood. Limited postmortem evidence has proved neural (neuron and oligodendrocyte) apoptosis, inflammatory responses, glial scar formation, neuronal degeneration/loss and axonal demyelination as the most prominent pathophysiological features for CSM [19]. In the present study, we have undertaken an investigation of the NVU changes along the natural course of SFR. We found that the compensatory change of NVU, in particular in the adjacent level of the compressive epicenter, was involved in functional recovery in experimental CSM model. This implies the effect of NVU compensation from the adjacent level of the myelopathic cervical cord on functional recovery.

The primary neuropathological features of CSM, which were well documented in postmortem CSM patient and clinically relevant models of CSM, include degeneration and loss of neuron in the ventral horn and loss of axon and demyelination in the posterior and lateral column of WM [4,19,20,21]. Agreeing with previous studies, the present study found significant loss of neurons and axons in the compressive epicenter but not in the adjacent level. Neuronal and axonal degeneration were verified by pathological and ultrastructural examination in the compressive epicenter, as well as with mild degenerative change in the adjacent level. Although neuronal and axonal degenerative change existed throughout the long-term compressive condition, it has to be noted that synaptic compensatory changes, in particular, such as increased synaptic vesicle and reactive synaptic mitochondria, were supposed to play an important role in compensatory pathomechanisms, which are responsible for spontaneous functional recovery. Our ultrastructural findings further strengthen the underlying pathomechanism that surgical decompression induced restoration of serotonergic fibers as one of endogenous mechanisms for promoting neurological recovery [22].

The NVU, mainly composed of the BSCB and neurons, is the essential functional unit of the CNS, which plays an important part in maintaining the integrity of the BSCB and regulating the permeability of the BSCB and microcirculation [9,23]. Structural disruptions of the BSCB commonly involve endothelial dysfunction/injury, tight junction breakdown, basement membrane corrosion and astrocytic endfeet detachment from the abluminal wall of capillaries [16]. The disappearance of EBA-immunoreactivity was reported to be correlated with tissues edema, inflammatory condition and astrocytic swelling [24,25]. On the contrary, the reappearance may be associated with restoration of the BSCB property benefitting from the surrounding astrocytic endfeet and compensatory proliferation of endothelial cells [26]. Loss of EBA immunoreactivity was documented in a period of approximately 2 weeks in the injury focal point, as well as in the adjacent microvessels in acute brain injury [24], while gradual restoration of EBA immunoreactivity was also observed within 3–4 weeks [18], which confirms that the endothelium, as one of the essential components of the NVU, plays a vital role in the pathophysiological process of CNS injury and recovery [12]. Further studies described significant decrement of immunostaining intensity, vascular profile number and area in the contusion and pericontusional site of the traumatic brain cortical, providing a quantitative method for exploring the microvascular alterations [25,27]. However, the correlation between morphological and quantitative change of EBA-positive microvessels and neurological functions was not carried out in these studies. In a traumatic SCI model, an earlier reappearance of EBA-immunoreactivity was found at the compressive site after 9 days thoracic cord compression [26]. However, the pathophysiological mechanism underlying reappearance of EBA-positive *microvessels* and the corresponding functional recovery remains unclear. In contrast with traumatic CSN injury, our aforementioned findings exhibited long-term loss of EBA immunoreactivity and reduction of RVPA in the compressive epicenter. The morphological characteristics of the EBA immunolabeling microvessel were quite consistent with the ultrastructural features by TEM examination. Furthermore, outstanding restoration of such changes was found in the adjacent level. The quantitative findings, to some extent, indicated dysfunction of microcirculation in the compressive site, while the compensatory change of the endothelial barrier in the adjacent level was an important compensatory change for improving microcirculation and thus inducing functional recovery. The pathophysiological change different from traumatic or acute CNS injury should be responsible for the unique pathogenic mechanism that persistent cord compression poses as long-term oppression to the microvessel [3,4]. In consequence, these findings indicated that compensatory restoration of endothelial barrier function and microcirculation may be one of the endogenous driving forces for inducing SFR in a CSM model.

Our TEM examination further proved that remarked vacuolation of endothelial cells which increased the defect of integrity and fuzzy structure of basement membrane were the ultrastructural characteristics of microvascular destruction in the compressive epicenter. Meanwhile, proliferation of astrocytic endfeet was more apparent in the adjacent level than that in the compressive epicenter. Vascular events, in particular, disruption of the BSCB, are usually the early pathophysiological change in SCI [28]. In the natural course of CSM, progressive chronic compression, inducing ischemia and subsequent disruption of BSCB, are considered to trigger a cascade of pathophysiological processes, including oxidative stress damage, neuroinflammation and neurotoxicity impairment, which finally lead to neuronal degeneration/apoptosis and axonal demyelination [29]. The latest research has proved that increased permeability and disruption of the BSCB are presented in patients with degenerative cervical myelopathy, of which damage magnitude was correlated with the severity of neurological symptoms [2]. In this study, we further validated that functional disruption of BSCB integrity was synchronized with neurological deficit in the early stage from the 1st day to 7th day of cord compression. From the 14th day to the observation termination period at the 2nd month, gradual SFR was documented with functional restoration of BSCB integrity. However, at the observation termination, EBA-immunoreactive microvessels showed a significant loss of immunoreactivity and decrement of RVPA in the compressive epicenter, which suggested persistent endothelial barrier dysfunction due to long-term compression in the compressive focal point. Meanwhile, in the adjacent level, an obvious restoration of immunoreactivity and increment of RVPA was seen, especially in the VH of GM. A similar chronic vascular change documented in a rare postmortem CSM patient study, which was characterized by abnormal thickening of the vasculature, might suggest that compensatory vasculature responds in long-term compressive conditions [19].

In the present study, conspicuous accumulation of gemistocytic astrocytes with proliferated elongated processes was identified in the compressive epicenter, which was different from the glial scar formation in acute SCI [30]. The number of astrocytes, oligodendrocytes and microglial cells increased significantly in the compressive epicenter compared to that in the control group. Reactive proliferation and aggregation change of neuroglial cells were related more mildly in the adjacent level, while drastic increment of microglial cells can be seen both in the compressive epicenter and the adjacent level. A growing number of studies have focused on the pathophysiological responses of neuroglial cells, which are supposed to be involved in natural process of CSM and the pathophysiological process of neurorestoration. Decreased myelin content with dysfunction of spinal cord conduction has been verified in a CSM patient [31]. Since oligodendrocytes are the key neuroglial cells responsible to the production of myelin sheaths and the self-repairing capability of myelin, apoptosis of oligodendrocytes has been implicated as one of essential pathomechanisms in CSM [19,32,33,34,35]. However, on the contrary, one of most recent studies has found that spontaneous locomotor recovery was not affected when the axons failed to remyelinate, which suggests that oligodendrocyte remyelination is non-essential for spontaneous recovery [36]. This conclusion may partially explain why spontaneous recovery occurred regardless of the existence of long-term axonal demyelination in the compressive epicenter in the present study. Therefore, whether axonal remyelination and subsequent functional recovery are ascribed to conspicuous increment of oligodendrocytes in the epicenter and adjacent level needs further validation.

Ultrastructural pathological lesions of astrocytes were documented in autopsied CSM patients [37]. It has been widely believed that overreactive astrocytes and astrocytic scars were considered as detrimental factors that prevent axonal regrowth [38,39]. Nevertheless, a recent study broke the prevailing view and drew the conclusion that astrocytic scar formation is beneficial for axonal regeneration [40]. Similarly, during the spontaneous recovery process, large number of activated astrocytes characterized by long thorn-shape processes presented in the epicenter as well as the adjacent level, which would prompt us to revisit the role of astrocytes in chronic compressive SCI.

The BSCB, neurons and neuroglial cells are the indispensable components of NVU, and are closely interrelated with each other structurally and functionally, thus, maintaining homeostasis of the CNS. In this present study, The Matrix Scatter Plot disclosed a moderate correlation between astrocyte number and vascular number/vascular area, as well as number of astrocytes and oligodendrocytes. Proliferated astrocytic endfeet also appeared in the epicenter and the adjacent level. Moreover, the disruption of the endothelial barrier was characterized by increased perivascular space and detached astrocyte endfeet in the compressive epicenter. In contrast, the perivascular astrocytic endfeet process was tightly wrapped with the abluminal vascular wall. Such a pathomorphological change of astrocyte–endothelial construction reacting to persistent compression, to some extent, indicated that the astrocytes struggle to maintain the integrity of the BSCB and prevent BSCB disruption spreading to the adjacent level [11,14]. The latest study further validates that astrocytes play an important part in regulating microcirculation in the brain and thus modulating the activity of neurons [41]. Homeostasis interaction among neuroglia, including oligodendrocytes, astrocytes and microglia enable the process of remyelination [42]. However, abnormal aggregation of perivascular oligodendrocyte precursor cells might disturb the astrocytic endfeet and endothelial tight junction integrity, which may disrupt the BBB, increase vascular permeability and induce inflammatory impairment [43].

In the adjacent level, we observed evident increased EBA-immunoreactive microvasculature number and area, with closely wrapping proliferated astrocytic endfeet and restoration of BSCB permeability, which may suggest that reactive astrocytes play an important role in maintaining the structural and functional integrity of the BSCB. The compensatory change of microvessels would contribute to rebalancing the microcirculation and promoting neuron survival and synaptic plasticity. An up-to-date study indicated that astrocytes play an important role in regulating cerebral perfusion and circulation condition to maintain sufficient bloody supply [41]. In addition, the number of astrocytes, oligodendrocytes and microglia increased significantly in the adjacent level, which may be associated with clearance of degeneration or damaged axon and axonal plasticity around the lesion site [42]. Nonetheless, in brief, the neurovascular compensatory responses, instead of consequential neural regeneration, originating from the spared cord adjacent to the mechanical compressive epicenter, play a crucial role in the natural process of spontaneous recovery of the CSM model.

The NVU has become a hotspot concept for investigating pathophysiological mechanisms of cerebrovascular disease (including ischemia cerebral infarction or hemorrhage) and neurodegeneration disorders, such as Alzheimer’s disease (AD), Parkinson’s disease (PD) and amyotrophic lateral sclerosis (ALS) [14]. A recent study revealed the ultrastructural damage to the NVU components in a rat model, including evident endothelium vacuoles, defects of tight junctions, an expansive basement membrane, swollen astrocytic endfeet and mitochondria [16]. This study also specifically pointed out compensatory ultrastructural changes of microvascular components, which are characterized by increased thickening of the endothelium, an expansive basement membrane, increased pericyte processing area and vessel coverage [16], while correlation analysis between compensatory change (particularly in the adjacent level) and spontaneous functional recovery in neuroglia and microvessels was not performed. To develop a drug therapeutic mechanism and effect in the NVU level, further attempts have established similar in vitro modeling of the NVU by co-culturing of endothelial cells, neural stem cells, astrocytes and pericytes, [44,45]. The latest therapeutic experimental study has obtained evidence on NVU component repairing and the consequent neurobehavior recovery in traumatic cervical SCI [46]. In consequence, the compensatory response to chronic compression in the NVU level may play a crucial role in spontaneous recovery in CSM.

The present study provided a general observation of the effect of NVU compensation changes during SFR of CSM. Findings of this study will be beneficial for new therapeutic strategy development of CSM. It should be noted that further study in cell biology and fundamental neurological studies are necessary to explain the underlying mechanisms of spontaneous recovery. It also helps to consider diagnostic and prognostic value in neurovascular measurement of the cervical spinal cord. A non-invasive way for spinal cord ultrastructure measurement of is one of the key issues in achieving accurate assessment and precise predictability of CSM. Advanced neuroimaging technology, such as diffusion tensor imaging (DTI) [47,48], could provide clear visualization and quantitative assessment of neural deficits in the spinal cord. It would be a promising tool for in vivo exploration of spinal cord ultrastructural features.

## 4. Materials and Methods

### 4.1. Animal Models

#### 4.1.1. Compression Material

A water-absorbing and progressive expanding synthetic polyurethane polymer sheet was used as the implanting compression material. The polyurethane polymer sheet was made of isocyanates and polyols (Guangzhou Fischer Chemical Co., Ltd., Guangzhou, China). A polymer sheet at size of 3 mm × 1 mm × 1 mm was cut and sterilized for implantation preparation. The compression sheet could absorb liquid in the epidural space to gradually expand its maximum volume and induced progressive compression to the spinal cord [49].

#### 4.1.2. Surgical Procedure

A total of 24 female adult Sprague–Dawley (SD) rats (250–300 g) were divided into compressive group for 2 months (n = 12) and control group (n = 12). All animal protocols were approved by the animal ethics committee of the affiliated hospital of Guangdong Medical University, Zhanjiang, Guangdong, China.

The animals were operated on under microscopy by a trained spine surgeon according to an established surgical protocol for implantation of water-absorbing materials [49]. In brief, the animals received general anesthesia with a mixture solution of 10% ketamine and 2% xylazine intraperitoneally. Preoperative location of the cervical spine was identified and marked. After the occipital and nuchal areas were shaved and sterilized, skin incision and thereafter blunt dissection were performed. The surrounding ligamentum flavum between C4 and C5 was exposed under microscopy. A thin polymer sheet was then carefully inserted into the left side of the rat spinal canal at the C5 level (Figure 1B). After implantation, the incision site was closed by layers. Then, the rats were recovered fully from the surgery on a heating bed and sent back to the cage freely for food and water. The polymer was expanding to reach maximum expansion in 24 h and remain at the maximal volume for 2 months, producing a chronic and persistent course of compression on the cervical spinal cord in the rats.

### 4.2. Neurological Function Assessment

Locomotor function was evaluated by using the 21-point Basso Beattie Bresnahan (BBB) scoring system [50]. Two spinal surgeons were invited to evaluate the locomotor function of the rats independently based on the scoring system. Scores were recorded at the timepoints of 1, 3, 7, 14, 21-days, 1-month and 2-month. The average score was calculated to depict the dynamic locomotor function of the modeling rats.

Sensory functional integrity of the cord among the model rats were evaluated by somatosensory evoked potential evaluations (SEP) using an established protocol (Zhang et al., 2009). A constant current stimulation (3.4 Hz square wave, 0.2 ms in duration, 0.3 ms time interval) was transmitted through the simulating electrode into the median nerve at the forelimb of the rat. The cortical SEP was recorded from the skull at Cz–Fz. The signal was amplified 2000 times by a band-pass filter between 10 and 2000 Hz (Zhuhai Yiruikeji Co., Ltd., Zhuhai, China). To obtain a good quality of the SEP signals, a total of 200 SEP responses was averaged for each trial. For synchronization observation of motor sensory function of the rats, SEP raw data were collected at the timepoint, consistent with BBB soring.

### 4.3. Tissue Preparation and Histopathology Examination

After the experiment, rats were sacrificed by overdose injection of intravenous sodium pentobarbital and perfused with 50 mL heparin saline solution through the ascending aorta thereafter with 300 mL formalin–picric solution (4% formaldehyde, 0.4% picric acid in 0.16 mol/L phosphate buffer, pH = 7.4). The whole cervical spinal cord was carefully harvested and fixed with 4% phosphate buffer liquid in formaldehyde solution for another 72 h. The compressive epicenter and adjacent level (5 mm caudal to the compressive epicenter) of the cord were cut, respectively, and embedded in paraffin. The specimens were continuously sectioned into transverse or sagittal slices of 4 μm thickness using a microtome.

The section specimens were stained with hematoxylin-eosin (H&E) and Luxol Fast blue (LFB, Sigma Chemical Co., St. Louis, MO, USA) staining. All images of the cords were acquired by a microscopic imaging system (FV-1000, Olympus, Japan). H&E staining was used to verify neuronal and axonal degeneration. All large motor neurons with clearly delineated centrally located nuclei and abundant Nissl substance within the perikarya were identified and counted in the ventral horn of the GM at ×10 view among all model rats using Image J 1.47 V. LFB was employed to stain the neurokeratin of myelinated fibers mainly in the WM of the spinal cord. The blue color intensity indicated the content of myelin.

### 4.4. Immunohistochemistry

After deparaffinating, the sections were treated with a 3% solution of hydrogen peroxide (H_2_O_2_) in 95% methanol (MeOH) to block endogenous peroxidase activity. The sections were rinsed in 0.01 M PBS (3 × 5 min). Then, the tissues were treated with 1:100 trypsin solution and 10 mM warm citrate buffer to unmask hidden antigens. The sections were rinsed in 0.01 MPBS solution (3 × 5 min) and the tissue was incubated for 30 min in 1:10 solution of normal goat serum in PBS at 37 °C. The solution was removed and the tissue was incubated in the dilutions of monoclonal Anti-Rat Blood–Brain Barrier antibody (Covance, Cat.No.SMI-71R, 1:1000), Anti-OLIG2 antibody (Bioss, bs-11194R, 1:200), Anti-AIF1/Iba1 antibody (Bioss, bs-1363R, 1:200) and Anti-GFAP antibody (Bioss, bs-0199R, 1:400) at 4 °C overnight. The primary antibody substituted for PBS was set for the negative control. The sections were incubated in the biotinylated secondary antibody, i.e., 1:20 goat anti-mouse IgG (Vector Laboratories, Inc., Cat. No. BP-9200) and goat anti-rabbit IgG (ZSGB-BIO, SP9001) for 1 h at 37 °C, respectively, rinsed in PBS (3 × 5 min) then added to an avidin–biotin–peroxidase complex (Elite^®^ ABC reagent, Cat. No.PK-7100) for 30 min. We used 3,3-diaminobenzidine (DAB) kit (Vector Laboratories, Inc., Cat. No. SK-4100) as visualization chromogen according to the manufacturer’s instructions. The tissues were incubated with DAB substrate for 30 s to 4 min. For quantitative analysis of neuroglial cells and their correlation with microvessels, nuclei of oligodendrocytes, astrocytes and microglial cells were counterstained with Mayer’s hematoxylin. The counterstain recipe can give clear and sharp nucleus staining with little background. After being dehydrated in a gradient of ethanol and cleared with xylene, the sections were cover-slipped using DPX Mountant and dried for microscopic observation.

### 4.5. Evan Blue Extravasation Measurement

For quantitative and qualitative analysis of the BSCB dysfunction severity, the contents of the extravasating EB and fluorescence were detected. The optical density (OD) value was measured by a spectrophotometer at 620 nm wavelength. The standard curve was drawn and the corresponding linear regression equation was calculated according to the testing result of EB solution at different concentrations (6, 3, 1.5, 0.75, 0.35, 0.175 μg/mL). A 2% EB solution (10 mL/kg; Sigma) was slowly intravenously administered through the vena caudalis. The solution was allowed 24 h for microcirculation and sufficient extravasation into the spinal cord parenchyma in vivo. After satisfactory anesthesia, the rats were transcardially perfused with 500 mL/kg saline. The intact cervical spinal cord was harvested and weighed, then kept in 3 mL methanamide solution for 72 h (37 °C). The cord tissues were centrifuged at 1500 r/min for 15 min. The supernatant was obtained to measure the OD value at 620 nm and the EB concentration can be calculated according to the standard cure. The EB content was calculated based on the following equation:EB(μg/g) = [EB concentration ((μg/mL) ×3 mL (methanamide))/weight of spinal cord tissue (g)

On the other side, the spinal cord tissue was frozen in liquid nitrogen and then embedded in OCT. Slices at 20 μm were sectioned and visualized using a confocal laser scan microscope system (Germany Leica, TCS SPS II; Olympus OLS-3100).

### 4.6. Transmission Electron Microscopy Investigation

Rats underwent satisfactory euthanasia with sodium pentobarbital solution. Transcardiac perfusion was not carried out in order to preserve the ultrastructure of spinal cord capillaries. The cervical spines of the rats were harvested and fixed in 4% paraformaldehyde (PFA) in 0.1 M PBS solution (PH 7.0) for 24 h at 4 °C. Subsequently, careful anatomical separation was performed to obtain an integral cervical spinal cord. Sagittal spinal cord fragments (3 mm × 1 mm × 1 mm) from the posterior funiculus of WM and GM were separated and then fixed with 2.5% glutaraldehyde in phosphate buffer (pH7.0) overnight. The cord specimens were washed three times in the PBS for 15 min and then postfixed with 1% OsO4 in PBS for 1h. After dehydration by a graded series of ethanol, specimens were embedded in Epon medium and heated at 70 °C for about 9 h. Finally, the specimen sections at 90 nm were stained by uranyl acetate and alkaline lead citrate, respectively, for 15 min and observed in TEM (JEOL, JEM-1400). Images acquired from imaging system (OLYMPUS DP74) were used to investigate ultrastructural features of neuro-vascular cytohistology.

For quantitative analysis of the ultrastructure of the microvessel and the neuron, as well as the neuroglial cells, micrographs (4008 × 2672 pixels, 24.559 × 16.373 μm, 6.137 nm/pixel) of TEM were acquired and analyzed using Image J 1.47V. The analyzing protocol was based on a previous ultrastructural study [16].

### 4.7. Quantitative Analysis of EBA-Immunopositive Microvessel and Neural Cells

On each anatomic region of the whole spinal cord, such as VH and DH of GM, AF, PF and LF of WM, 5 fields of ×20 view (294 × 236 μm^2^) image were acquired with constant imaging parameter by a microscopic imaging system (FV-1000, Olympus, Japan). For quantitative analysis of blood vessels, the selected regional EBA immunostained vascular profile per filed was outlined by Image J 1.47 V. Then, the RVPN and RVPA per filed were automatically computed. Average RVPN and average RVPA were calculated in different anatomic regions. Percentages of the average RVPN and RVPA were also calculated to explore the distribution of vasculature in the spinal cord.

To quantitatively analyze the neuroglial cells, two practiced pathology technicians were invited to classify and count the number of oligodendrocytes, astrocytes and microglial cells independently on ×20 images. The types of neuroglia cells were primarily distinguished based on morphologic features of the nucleus, which were presented in the EBA immunostaining section. The quantity of neuroglia cells was counted in different anatomic regions. Then, mean numbers of oligodendrocytes, astrocytes and microglial cells were computed by averaging the results from the pathology technicians.

### 4.8. Statistical Analysis

For intra-group comparison of RVPN, RVPA and neuroglial cell number in different anatomic regions, an independent t-test was performed by using SPSS 16.0 (SPSS Inc., Chicago, IL, USA). For inter-group comparison among control group, compressive epicenter and adjacent level, a one-way analysis of variance (ANOVA) and post hoc test were performed. Matrix Scatter Plot and Linear Correlation analysis were used to explore the relationship between RVPN, RVPA and neuroglia cells. Data are presented as mean values ± SEM and *p* < 0.05 was considered statistic significant.

## 5. Conclusions

Limitations should be considering when addressing the results in the present study. Firstly, the dissimilarity of life span and nervous system between rodents and humans would produce different pathophysiological characteristics under chronic, persistent spinal cord compression. Secondly, the water-absorbing expanding polymer sheet creates post-lateral compression in the present CSM rat model, which is not the most frequent compressive site of CSM in humans. In addition, temporal investigations on the pathological and pathophysiological changes and molecular biological alterations are needed to further elucidate the underlying mechanisms of spontaneous recovery. In conclusion, the results may initially present the identification of pathomechanisms of neurovascular compensation that induce spontaneous restoration, and which originated from the adjacent spared cord, which indicate greater potential of regeneration and plasticity in CSM. The understanding of the natural restoration process will help us in establishing a theoretical foundation in surgical decision-making and surgical timing. Additionally, the potential of neurovasculature regarding a neural regeneration strategy deserves more attention in CSM. Finally, it is necessary to evaluate the spontaneous recovery impact when determining any experimental therapeutic efficacy including but not limit to pharmacotherapy.

## Figures and Tables

**Figure 1 ijms-24-03408-f001:**
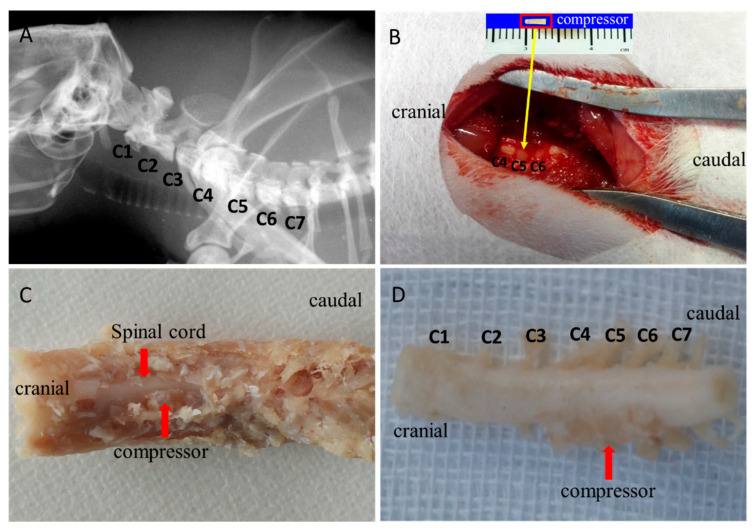
Establishment of experimental CSM model. (**A**) Preoperative spinal localization by X-ray fluoroscopy. (**B**) C4 lamina removing and polymer sheet implantation into C5 epidural space. (**C**) Ex vivo spinal specimen with compressor. (**D**) Expanded compressor identified at C5–C6 cord level.

**Figure 2 ijms-24-03408-f002:**
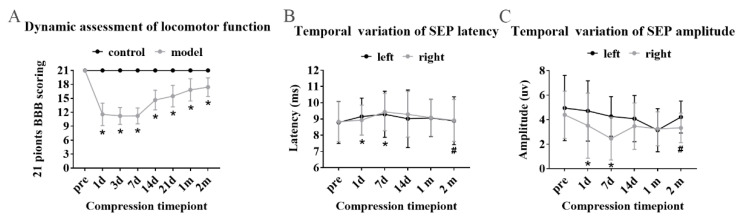
SFR was confirmed in modeling rats. (**A**) Obvious locomotor function recovery since 7th day showed by BBB scoring. (**B**,**C**) Shortening latency and increasing amplitude of SEP was identified from 7th day. “*” Compared with preoperative, “#” compared with 7th day (*p* < 0.05).

**Figure 3 ijms-24-03408-f003:**
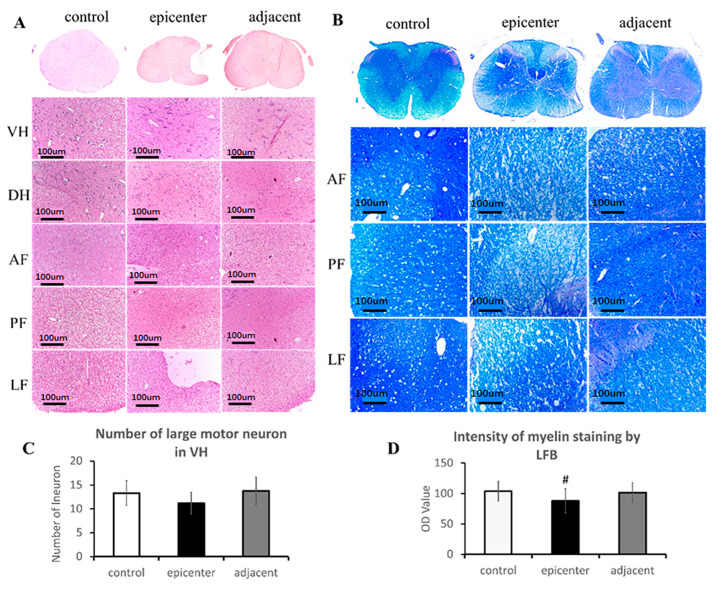
Histopathological evidence of neuronal and axonal degeneration. (**A**) Chronic compression caused neuronal degeneration in VH and DH and neural fiber disorganization in AF, PF and LF (HE staining). (**B**) Chronic compression caused axonal demyelination in AF, PF and LF (LFB staining). Evident vacuolated change of neural fiber with significant reduction of myelin staining intensity demonstrated axonal demyelination in the compressive epicenter. (**C**) No significant difference in large motor neuron number was found among the three groups. (**D**) Milder axonal demyelination was also seen in the adjacent level and no significant difference of myelin staining intensity was found compared with control group. Note: “VH”, ventral horn; “DH”, dorsal horn; “AF”, anterior funiculus; “PF”, post funiculus; “LF”, lateral funiculus; “#”, different from the control group (*p* < 0.05).

**Figure 4 ijms-24-03408-f004:**
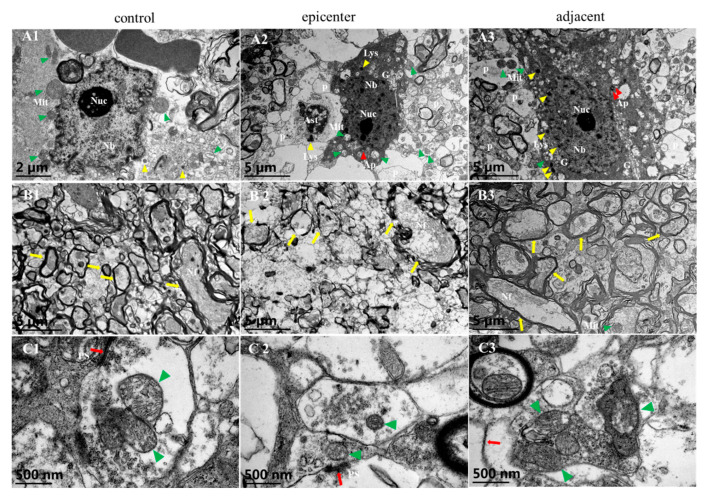
Ultrastructural evidence of neuronal degeneration and axonal demyelination. Large motoneuron and its axon with axonal terminal in control group (**A1**,**B1**,**C1**), compressive epicenter (**A2**,**B2**,**C2**) and adjacent level (**A3**,**B3**,**C3**). (**A1**) Even distribution of Nissl body (Nb) in nucleus (Nuc), abundance of mitochondria (Mit) with distinguished cristae and several process (P). (**B1**) Clearly identified structure of the myelin and axon. (**C1**) Synapse and its mitochondria and post synapse (PS). (**A2**) Increased lysosome (Lys), autophagosome (Ap) and disappearance of mitochondrial cristae implied neuronal degeneration and death. (**B2**) Thinner layer of myelin and disorganization or destruction of axon. (**C2**) Abnormal synapse with blurred mitochondrial cristae and loss of synaptic vesicle. (**A3**) Amelioration change of neuron in adjacent level. (**B3**) Thicker myelin sheath but with disorganized myelin laminae and neurofilament (Nf). (**C3**) Similar appearance of synapse as that in the control group but with increased and aggregated synaptic vesicles.

**Figure 5 ijms-24-03408-f005:**
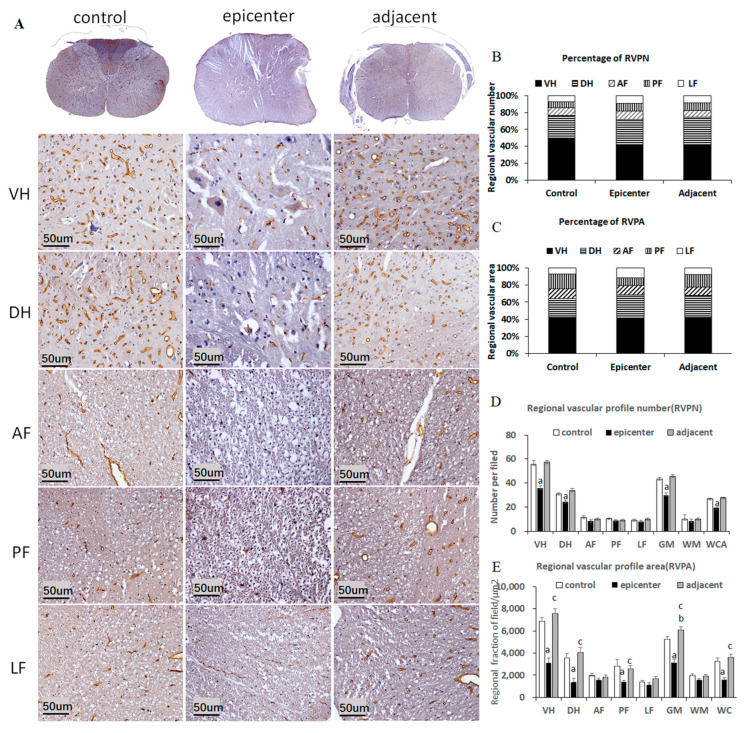
EBA immunoreactivity and microvascular anatomical distribution. (**A**) EBA immunoreactivity in different anatomic regions and groups. RVPN (**B**)/RVPA (**C**) anatomic distribution. The largest proportion of blood vessels is located in GM, especially in VH. (**D**,**E**) Quantitative analysis of blood vessels in different anatomic regions. The RVPN/RVPA decreased significantly in the VH and DH of the compressive epicenter, whereas increased they in the adjacent level. Data are presented as mean values ± SE for standardized ×20 microscopic fields (0.0694 mm²); “a, b”, different from control group; “c”, adjacent level different from the compressive epicenter (*p* < 0.05, repeated measures ANOVA followed by Bonferroni tests). Note: “VH”, ventral horn; “DH”, dorsal horn; “AF”, anterior funiculus; “PF”, post funiculus; “LF”, lateral funiculus.

**Figure 6 ijms-24-03408-f006:**
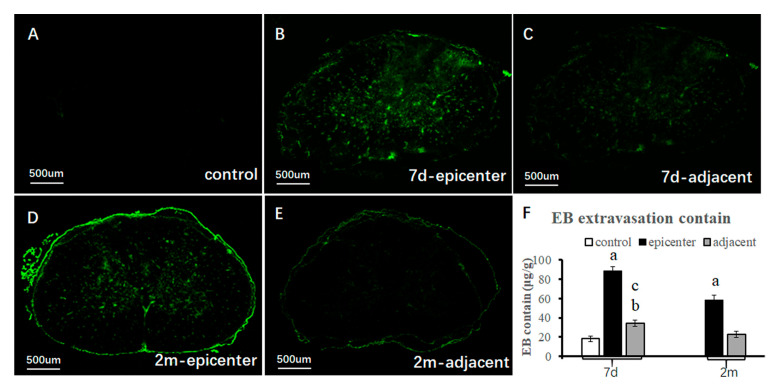
Disruption of BSCB integrity. (**A**) Negative EB fluorescence in control group. (**B**) EB fluorescence widely spread into the GM of the compressive epicenter at the 7th day. (**C**) Obvious decreased EB fluorescence in adjacent level at the 7th day. (**D**) Decreased EB fluorescence in the compressive epicenter at the 2nd month. (**E**) Significantly decreased EB fluorescence in adjacent level at 2nd month. (**F**) EB extravasation content increased significantly in the epicenter and adjacent level at the 7th day, while it decreased at the 2nd month. No significant difference was found between the adjacent level at 2nd month and the control group. “a,b”. compared with control group; “c”, compressive epicenter compared with adjacent level (*p* < 0.01).

**Figure 7 ijms-24-03408-f007:**
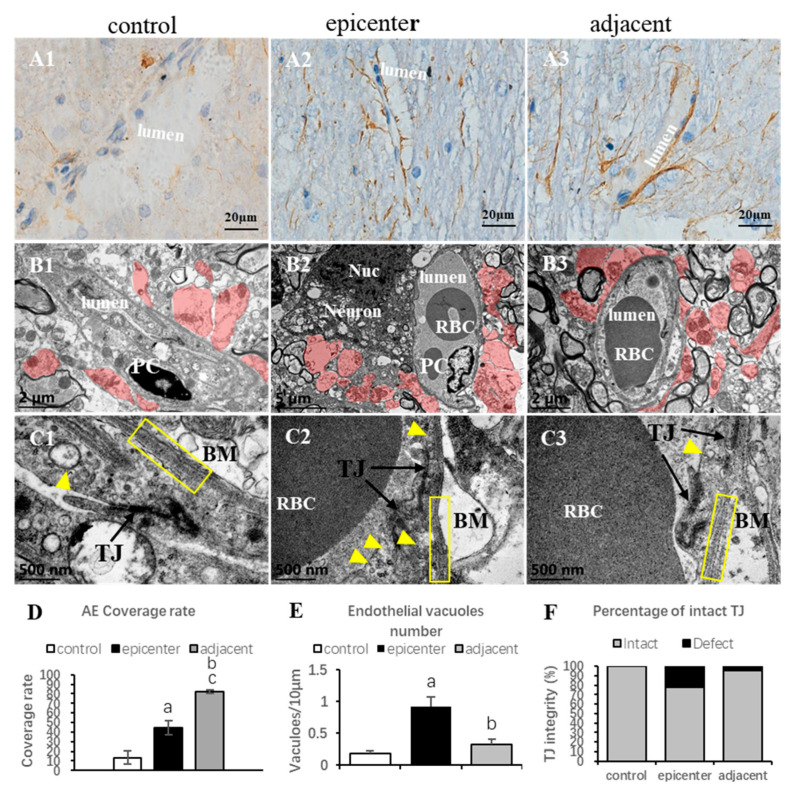
Microvascular compensation in the adjacent level. (**A1**, **A2**, **A3**) Perivascular astrocyte endfeet. (**A1**) Sparse astrocyte endfeet in the VH of the control group. (**A2**) Astrocyte endfeet detach from the endothelial cells of *microvessels* in the compressive epicenter. (**A3**) Proliferated astrocyte endfeet in the adjacent level. (**B1**–**B3**) Neurovascular ultrastructure. (**B1**,**C1**) Normal microvascular components including distinct endothelium, high electronic density TJ and PC) nucleus, clear lamina of BM and surrounded perivascular astrocytic endfeet (red shadow) in control group. Homogeneous electron density. (**B2**,**C2**) NVU degeneration in the compressive epicenter, including collapsed microvascular wall, disorganized BM and loss of TJ integrity, neuron degeneration and apoptosis, formation of phagolysosomes and autophagic vacuoles. (**B3**,**C3**) Restoration of microvascular microstructure. (**D**) Significant increase of astrocytic endfeet both in the compressive epicenter and adjacent level. (**E**) Significant decrease of vacuoles number in endothelial in adjacent level. (**F**) Restoration of TJ integrity in adjacent level. PC (pericyte), Nuc (nucleus), TJ (tight junction), BM (basement membrane), RBC (red blood cell); “a,b”. compared with control group; “c”, adjacent level compared with the compressive epicenter (*p* < 0.05).

**Figure 8 ijms-24-03408-f008:**
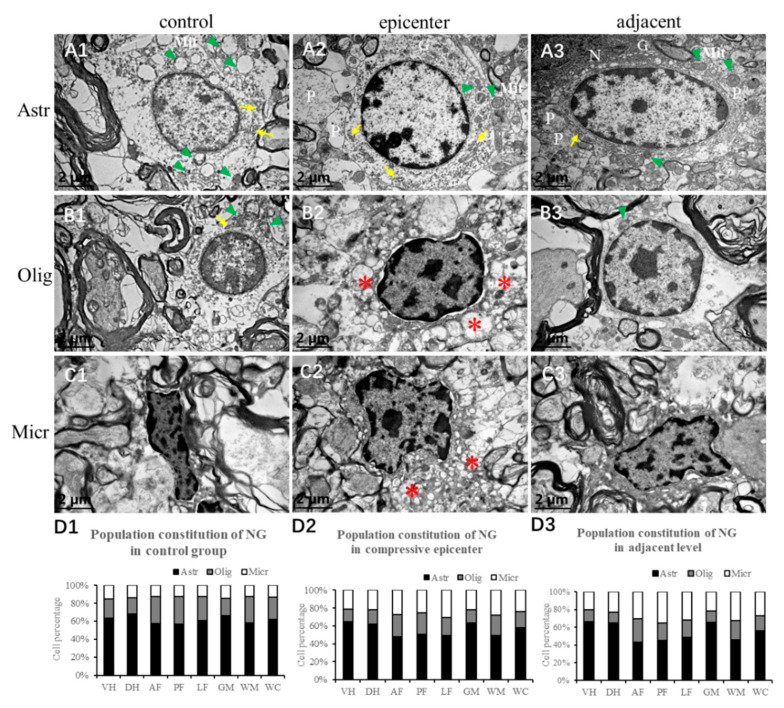
Morphology of neuroglial cells by TEM examination. (**A1**–**A3**) Astrocytes (Ast) exhibited even distribution and low-density karyoplasm with abundant cytoplasm and cytoplasmic organelles such as granular endoplasmic reticulum (yellow arrow), Golgi complexes (G), ribosomes and mitochondria (green arrowhead), as well Nemours processes (P); (**B1**–**B3**) Oligodendrocytes (Olig) were smaller than the astrocytes with dense clumped chromatin aggregated eccentrically around the rim of the karyolemma. Numerous phagocytosed vacuolations (*) were identified; (**C1**–**C3**) Microglial cells (Micr) appeared to have irregular morphology and with clumped chromatin. Numerous phagocytosed vacuolations (*) were seen in the compressive epicenter. (**D1**–**D3**) Population constitution of neuroglial cells (NG) in different anatomic regions.

**Figure 9 ijms-24-03408-f009:**
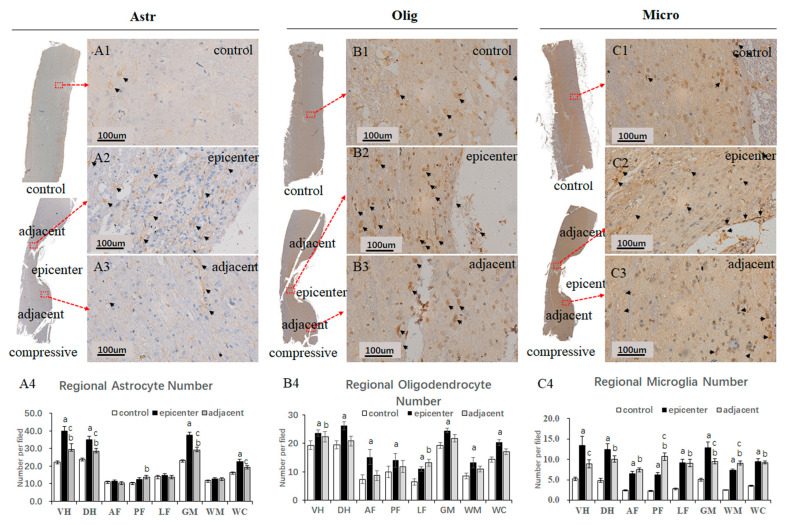
Neuroglial cells activation after chronic compression. Anti-GFAP, Anti-OLIG2 and Anti-AIF1/Iba1 antibody labeled astrocytes (**A1**–**A3**), oligodendrocytes (**B1**–**B3**) and microglial cells (**C1**–**C3**) in the control group, compressive epicenter and adjacent level, respectively. A large number of reactive astrocytes (**A2**, black arrowhead), oligodendrocytes (**B2**) and activated microglial cells (**C2**) were seen in the compressive epicenter. (**A3**) Milder increased activation of astrocytes (**A3**), oligodendrocytes (**B3**) and activated microglial cells (**C3**) were also seen in the adjacent level. Regional quantitative analysis of results demonstrated an increased number of astrocytes (**A4**), oligodendrocytes (**B4**) and microglial cells (**C4**) after compression, especially in the GM of the compressive epicenter, as well as the adjacent level. Regional number per field are presented as mean values ± SE for standardized ×20 microscopic fields (μm²). Note: a,b. compressive epicenter/adjacent level compared to control group; c, compressive epicenter compared with adjacent level (*p* < 0.01).

**Figure 10 ijms-24-03408-f010:**
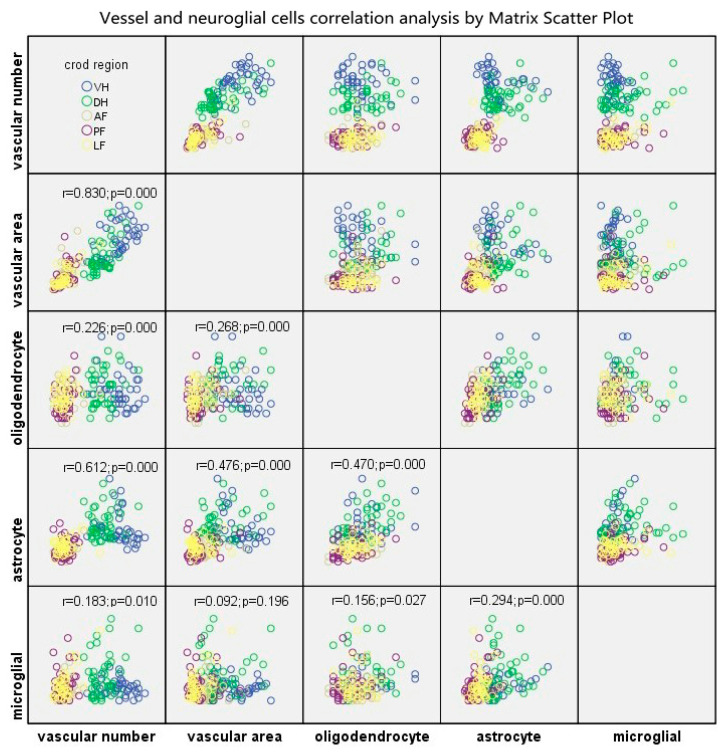
Interrelation among microvascular number/area and neuroglial cells including oligodendrocytes, astrocytes and microglial cells. Note: “VH”, ventral horn; “DH”, dorsal horn; “AF”, anterior funiculus; “PF”, post funiculus; “LF”, lateral funiculus.

## Data Availability

Not applicable.

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
