# Peer review of "Neurovascular Unit Compensation from Adjacent Level May Contribute to Spontaneous Functional Recovery in Experimental Cervical Spondylotic Myelopathy"

_ijms, 2023, doi:10.3390/ijms24043408_

Round 1
Reviewer 1 Report
I think this work is very interesting, but in order to understand it properly, the Material and Methods section should be placed before the Results sectionAuthor Response
Thank you for your encouraging comments. According to the format of the journal, the Results section should be placed before the Material and Methods section. However, we found the reason misleading in our original manuscript. The content of original manuscript did not consider the order of sections. Therefor we revised the manuscript, including: introduce every abbreviation and the definition of measurement at the first appearance. We also make some modification to let the reader understandable in such order of manuscript.
Reviewer 2 Report
An interesting publication on the issues of progression and remission of cervical spondylotic myelopathy (CSM) and their unpredictability due to the ambiguous pathomchanism. Quite extensively and at the same time concisely presented topic. However, I would like to point out the shortcomings in showing the methodology of monitoring these processes in a non-invasive way in order to provide more data, which would perhaps help to achieve greater predictability.
In the past, MRI techniques and DTI in particular, whether in vitro or in vivo, have been used to try to figure out where the changes are going. I think you should consider these issues:
- Visualization of the extent of damage in a rat spinal cord by DTI.
- Quantitative assessment of injury in rat spinal cords in vivo by MRI and DTI.
In addition, you can also learn about BSD-DTI techniques that increase the accuracy of DTI, and thus the chance for a proper diagnosis or understanding of a given mechanism. A discussion of these topics is advisable.
I also suggest improving the descriptions of the x, y axes, graphs, they are illegible in places.
Author Response
We agreed with your points and appreciate your suggestion on using DTI.
As you mentioned, DTI techniques was able to be used for detecting the extent of damage and quantitative analyzing non-invasively. In order to improve the current research, we have carried out relevant research in the planning follow-up research. DTI evaluation of neurovascular impairment in spinal cord has been discussed in the text. This issue was added to the last paragraph of discussion section as ‘It also helps to consider diagnostic and prognostic value on neurovascular measurement of cervical spinal cord. A non-invasive way for spinal cord ultrastructure measurement of is one of key issues to achieve accurate assessment and precise predictability of CSM. The advanced neuroimaging technology like diffusion tensor imaging (DTI) [47, 48] could provide clear visualization and quantitative assessment of neural deficits in the spinal cord. It would be a promising tool for in-vivo exploration of spinal cord ultrastructure features.’
The descriptions of the x, y axes, graphs have been improved.
Reviewer 3 Report
We carefully reviewed the present manuscript submitted for publication in the International Journal of Molecular Sciences.
The authors present an article on spontaneous recovery in cervical spondylotic myelopathy and the role of the neurovascular unit compensation. By grafting an expandable water-absorbing polyurethane polymer at C5 level have been analyzed neurological function up to 2 months in 24 rats.
The study presents an initial identification of the recognition of pathomechanism of neurovascular compensation, but does not fully explain the underlying mechanism of spontaneous recovery.
The topic is interesting and the scientific analysis is fine; the results confirm someone already known data, but doesn't add any new insight to the pertinent literature. Please accept the following criticisms:
• The structure of the article is unordered. For exemple, materials and methods should be exposed before the results.
• Figures don't always have a clear description associated with them and Figure 2 does not appear to be present
• It is too long
Author Response
We thank you very much for your valuable comments. According to the format requirement of the journal, Results section should be placed before the Material and Methods section. We have modified the legends that were not clearly described. Figure 2 was hidden under Figure 1 because of article editing, which has now been corrected. We made a major revision of the manuscript. The article has been shortened from 30 pages to 20 pages.
Reviewer 4 Report
The authors present an interesting study describing changes of the neurovascular unit in the adjacent level to compressive epicenters in cervical spondylotic myelopathy. The endpoint (spontaneous functional recovery) was investigated in an experimental rat model. Restoration of the blood spinal cord barrier permeability and increase of the regional vascular profile area with proliferated astrocytic endfeet, neuron surviving and synaptic plasticity were confirmed in the adjacent level. All in all, I read the manuscript with great interest and the manuscript is well written. In the following I have only some minor remarks:
- The present version includes some typing errors (e.g., line 111: From14-day…; line 131: FIGURE3C…). The authors should revise this
- Discussion: What is the clinical implication of the present manuscript? Are there any implications regarding future drug developments? How do we have to monitor patients with a cervical spondylotic myelopathy to estimate their prognosis? Is there a potential role of the reorganization of the motor area and excitability? (e.g., PMID: 29165642)
I think the manuscript can be considered for publication after a sufficient revision clarifying those remarks.
Author Response
The typing errors including in the original manuscript have been corrected.
The clinical implication including development of ‘new therapeutic strategy’ and ‘diagnostic and prognostic value of new tools to evaluate spinal cord ultrastructure’.
We are sorry that there was no sufficient results in this manuscript to elucidate the potential role of the reorganization of the motor area and excitability.
Round 2
Reviewer 3 Report
We are satisfied to note that the critical issues we have reported have been accepted.
We have carefully reviewed the new version of the article: we have noticed the improvement of the grammar and form, integration of the missing images and the revision of the citations. Also, abbreviations are used better, even in image descriptions. Of course, we confirm that the article doesn't add any new insight to the pertinent literature. Unfortunately, we still note that the article is poorly organized and complex in drafting.